# Augmented Flow Matching via Variance Reduction with Auxiliary Variables

## Abstract

Flow matching is a simulation-free approach that scalably generates an Ordinary Differential Equation (ODE), in which its path traverses between two different distributions. However, conventional flow matching relies on the training pairs drawn independently, inducing high variance that might slow down training process and degrade the performance upon training. To mitigate this, we propose *Augmented Flow Matching* (AFM), a simple yet efficient framework that can be ubiquitously applied to flow matching with slight modification to the models. We first find that when some auxiliary variables are correlated to the training data, then they contribute to variance reduction of the flow matching loss estimation, when used together with the training data pair. With this observation, we construct auxiliary variables that are correlated to the training pair, which is obtained by simple and effective linear operation from the input data. Finally, we show that with this simple modification on the training phase, we achieve the improved model flexibility and performance when the ODE is applied on the learned model.

## 1 Introduction

Deep generative modeling is considered as the problem of approximating the probability density and sampling from the distribution. As the deep learning framework emerged, normalizing flows (Papamakarios et al., 2021; Rezende & Mohamed, 2015; Papamakarios et al., 2017; Chen et al., 2019), a likelihood-based model developed from the variational autoencoder (Kingma & Welling, 2014), constructed an invertible mapping between a tractable distribution that is easy to be sampled to a complex distribution which is generally intractable. In spite of their successes, these models require invertible modules to work, which prevents them from catching up with the state-of-the-art performances that are achieved with Generative Adversarial Network (GAN) (Goodfellow et al., 2014) in generating high-quality examples. The continuous normalizing flow (Chen et al., 2018; Grathwohl et al., 2019) lifted those restrictions on invertible modules using neural ordinary differential equation (NODE), enabling the evaluation of free-form Jacobian on continuous-time ODE, in a memory-efficient way. However, they also have scalability issues, as they use the adjoint sensitivity method which runs a full simulation within the ODE trajectory in the training phase.

As a counterpart, diffusion models (Song et al., 2021; Ho et al., 2020; Sohl-Dickstein et al., 2015) first process the forward stochastic processes which move data to noise, then trained the reverse process by learning the score function of the data distribution. And Huang et al. (2021); Kingma et al. (2021); Song et al. (2021) showed that the forward and reverse stochastic process, represented by the Stochastic Differential Equation (SDE), has its equivalent ODE that induces the same evolution of probability density function, according to the Fokker-Planck equation. Inspired by this observation, flow matching Lipman et al. (2023); Liu et al. (2023) is a simulation-free method to learn the ODE drift in a similar manner to diffusion models. By doing this, one can not only generate data from noise, but also transfer the *source* distribution to *target* distributions.

However, training the ODE with flow matching suffers from high variance by randomized pairing between the source and target dataset. Pooladian et al. (2023); Tong et al. (2024); Song et al. (2023) generated the pair between those two distributions by constructing the Optimal Transport (OT) mapping between source and target data (Pooladian et al., 2023; Tong et al., 2024) or canonicalization via the rotation matrix (Song et al., 2023). However, the optimal transport approach is costly to be applied with larger minibatch size, as the complexity of matching algorithm, such as Hungarian al-

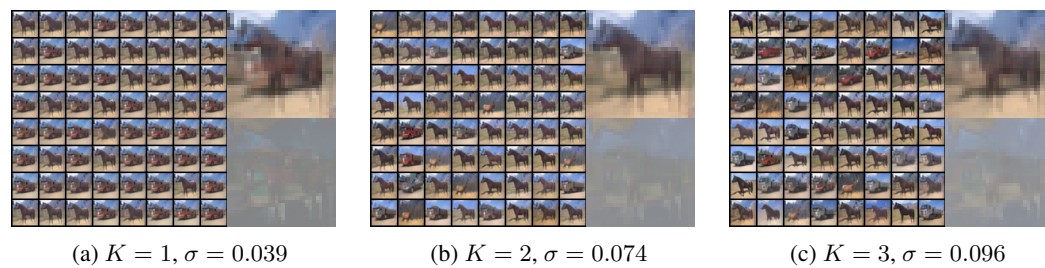

(a) $K = 1, \sigma = 0.039$      (b) $K = 2, \sigma = 0.074$      (c) $K = 3, \sigma = 0.096$

Figure 1: CIFAR-10 generated images. The images are generated with 50-step Euler solver, from the same initial data-space latents and varying augmented latents. Example images, their means, and the pixel-wise standard deviations are drawn in left, upper right, and lower right, respectively.

gorithm, explodes in the polynomial degree of the minibatch size. Furthermore, the OT approach is suboptimal for small minibatch size, that usually happens in the high-dimensional dataset in which the number of minibatch sizes is limited because of memory budget.

To resolve this issue, we first observed that the training variance of flow matching can be effectively mitigated by imposing a correlation between the training pairs. There have been a time-honored line of works in the variance reduction on Monte Carlo estimation. For instance, Rao-blackwellization (Rao, 1992; Blackwell, 1947) proposes that the conditional estimator for a given sufficient statistic always has reduced variance compared to the original estimator. Inspired by this, we first clarify that correlating the source and target distribution with augmented random variables effectively reduces the training variance. Then, we propose an simple and effective method, *Augmented Flow Matching* (AFM), that designs the random variables that augment a small number of dimensions in both source and target. Finally, we show that the flow matching with augmented random variables outperforms that of the naïve flow matching in a variety of datasets, both qualitatively and quantitatively. This approach can be plugged-in to other existing training methods that enhances efficiency, such as optimal transport or curvature-minimizing approach (Lee et al., 2023).

To summarize, our contributions are the following:

- Inspired by the statistical perspective, we first propose the variance reduction theorem in flow matching, such that the variance of the flow estimator reduces by introducing some auxiliary random variables as conditions, which allows for more effective training.

- Then, we propose a simple and powerful method that entangles the training data pair, by introducing the auxiliary variables. As this auxiliary variable is accessible in the training phase, we can successfully take advantage of the variance reduction which happens in the presence of those auxiliary variables. While a variety of choices are available for choosing the auxiliary variable, we chose the simplest form for universality: the linear multiplication between source and target data. These auxiliary variables are used together with the training data for training, augmenting the data dimension of the model.

- Finally, we verify that by training with these auxiliary variables, we achieve improved performance in various datasets, both in qualitative and quantitative ways, with sparse modifications on the baseline networks.

## 2 BACKGROUND: CONTINUOUS NORMALIZING FLOW AND FLOW MATCHING

In this section, we provide the background on continuous normalizing flow and the flow matching framework in Section 2.1 and Section 2.2, respectively.

### 2.1 CONTINUOUS NORMALIZING FLOWS

Normalizing flow (Rezende & Mohamed, 2015; Papamakarios et al., 2021) is designed to learn complex probability density functions, by the change of variables in the probability densities via invertible transformation. Normalizing flow enables sampling from a tractable, easy-to-sample prior

$K = 0$

$K = 2$

$K = 8$

$K = 20$

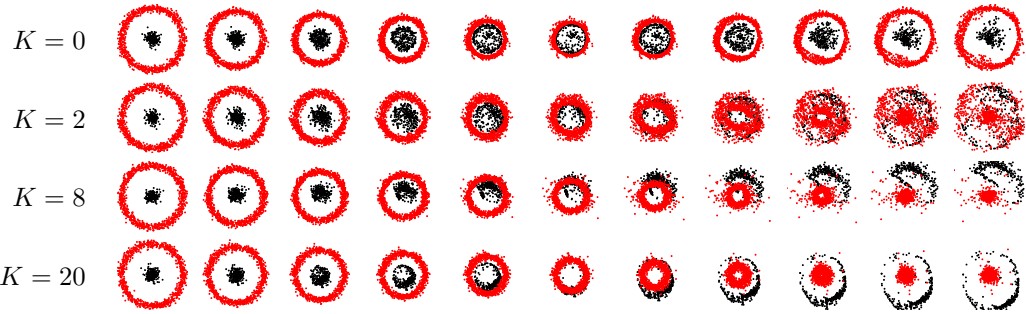

Figure 2: *Augmented flow matching* on 2D synthetic datasets. From time $t = 0$ (**left**) to $t = 1$ (**right**), **black** clouds are trained to go outside of the center, while **red** clouds are trained to get into the center. $K$ denotes the additional dimensions concatenated to the input, where auxiliary random variables lie. As $K$ gets higher, the learned dynamics become more flexible, making both red and black clouds possible to propagate. The detailed problem setting is described in Appendix A.

distribution (i.e., Gaussian distribution), to the complex function $f_\theta : \mathbb{R}^D \to \mathbb{R}^D$. For instance, suppose the tractable prior distribution be Gaussian, i.e., $\pi_0 \sim \mathcal{N}(0, I)$, and let the learned complex distribution be $p_\theta \in \mathcal{C}^1(\mathbb{R}^D)$, where $\mathcal{C}^1$ denotes to the continuously differentiable distribution over $\mathbb{R}^D$.

Then, this $p_\theta$ is obtained through the *push-forward* equation, (i.e., (generalized) change-of-variables)

$$p_\theta(x) = \mathcal{N}(f_\theta^{-1}(x)|0, I) \left| \det \left[ \frac{\partial f_\theta^{-1}(x)}{\partial x} \right] \right|. \tag{1}$$

Even though the normalizing flow provides a strong tool in designing the complex distribution from a simple distribution, it remains difficult to construct an invertible function $f_\theta$ (Chen et al., 2019), as equation 1 requires the Jacobian determinant of the inverse function of $f_\theta$. To mitigate this, Chen et al. (2018); Grathwohl et al. (2019) proposed the normalizing flows derived from the ODE. To form the ODE, the function $f_\theta$ is now parameterized from the *vector field* $v_\theta : \mathbb{R}^D \times [0, 1] \to \mathbb{R}^D$ recursively, as follows.

$$f_\theta(x) = g_\theta(x, 1), \quad g_\theta(x, t) = x + \int_0^t v_\theta(g_\theta(x_0, t'), t')\mathrm{d}t' \text{ with } x_0 \sim \pi_0, \tag{2}$$

which is reduced to $\mathrm{d}x_t = v_\theta(x_t, t)\mathrm{d}t$ with $x_t = g_\theta(x_0, t)$ if represented with differential form. Like in equation 1, $v_\theta$ generates the flow $\phi_t$ if it satisfies the push-forward equation

$$p_t = [\phi_t]_* \pi_0, \quad ([\phi_t]_* \pi_0)(x) = \pi_0(\phi_t^{-1}(x)) \left| \det \left[ \frac{\partial \phi_t^{-1}}{\partial x}(x) \right] \right|, \tag{3}$$

which enables $f_\theta$ to be invertible, without requiring the full Jacobian determinant of the inverse function $f_\theta^{-1}$, but just the inverse flow $\phi_t^{-1}$.

## 2.2 FLOW MATCHING

Despite these advantages of the Continuous Normalizing Flow (CNF), this requires the costly forward propagation of neural ODEs. To avoid this, Lipman et al. (2023); Liu et al. (2023) proposed the simulation-free (i.e., do not demand the full ODE computation) method called *flow matching*, which enables fast and scalable generation of ODE through light computational budget.

The flow matching objective directly regresses the vector field $v_\theta$ into the (true) vector field $u(x, t)$ as follows:

$$\mathcal{L}_{\text{FM}}(\theta) = \mathbb{E}_{t\sim[0,1], x_t\sim p_t(x_t)} \left[ \|v_\theta(x_t, t) - u(x_t, t)\|^2 \right] \tag{4}$$

where $p_t$ is defined in Equation 3. However, $u(x, t)$ is generally intractable, hence we instead use the following *conditional* flow matching (CFM) objective

$$\mathcal{L}_{\text{CFM}}(\theta) = \mathbb{E}_{t\sim[0,1], x_t\sim p_t(x_t|z), z\sim q(x_0, x_1)} \left[ \|v_\theta(x_t, t) - u(x_t, t|z)\|^2 \right], \tag{5}$$

---

**Algorithm 1** Training of Augmented Flow Matching

---

**Require:** Noise distribution $\pi(x_0)$, data distribution $p_1(x_1)$, number of augmented dimension $K$,
Number of channels $C$, Hyperparameters $\lambda_1, \lambda_2$. Neural network parameters $\theta$
Draw projection matrix $P \in \mathbb{R}^{K \times C}$, $P_{i,j} \sim \mathcal{N}(0, I)$ for all $i, j$.
Initialize parameters $\theta$ of neural network $\tilde{v}_\theta$.
**while** Not converged **do**
$\quad$ Draw $(x_0, x_1) \sim \pi(x_0) \times p_1(x_1)$, $t \in (0, 1)$.
$\quad$ Draw $y_0 \sim \mathcal{N}(0, I^C)$
$\quad$ $y_1 \leftarrow \lambda_1 y_0 + \lambda_2 P(x_0 + x_1)$
$\quad$ $\tilde{x}_t \leftarrow (1 - t)\tilde{x}_0 + t\tilde{x}_1, \tilde{x}_0 = [x_0, y_0]$ and $\tilde{x}_1 = [x_1, y_1]$
$\quad$ Update $\theta$ to minimize $\mathbb{E}_{\tilde{x}_t, t} \left[ \|\tilde{v}_\theta(\tilde{x}_t, t) - (\tilde{x}_1 - \tilde{x}_0)\|_2^2 \right]$
**end while**

---

with efficient formulation of $q(\cdot)$. For example, in Liu et al. (2023), the condition $z$ is independently drawn from the source and target distribution $z \sim q(x_0, x_1) = \pi_0(x_0)p_1(x_1)$. Lipman et al. (2023) showed that the gradient of $\mathcal{L}_{\text{FM}}(\theta)$ is equal to that of $\mathcal{L}_{\text{CFM}}(\theta)$; i.e., $\nabla_\theta \mathcal{L}_{\text{FM}}(\theta) = \nabla_\theta \mathcal{L}_{\text{CFM}}(\theta)$, which implies that the CFM objective can be utilized instead of the intractable FM objective for training $v_\theta(x, t)$ to simulate $u(x, t)$. Furthermore, to provide stability and training efficiency, Tong et al. (2024); Pooladian et al. (2023) used the optimal transport (OT) approach to find the coupling between the samples from $q(x_0)$ and $q(x_1)$ that minimizes the Wasserstein distance. Nevertheless, the following issues remain: it takes polynomial time to obtain the coupling between two different distributions via the optimal transport algorithm, which explodes in time when using a large number of minibatch sizes, and the coupling obtained from the minibatch coupling becomes uninformative in the entire dataset (Albergo et al., 2024).

## 3 AUGMENTED FLOW MATCHING

In this section, we propose our main method, *Augmented Flow Matching* (AFM), a simple yet efficient method that improves the training and sampling processes of flow matching with constructing auxiliary random variables that bootstraps training and is also used for sampling. First, in § 3.1, we justify how using other random variables, that are correlated to the training pair, improve the training process by reducing the training variance. Then, in § 3.2, we introduce the framework of using both the original pair and the generated pair into learning the flow matching model. Finally, in § 3.3, we propose an algorithm to generate the pair, where the source part is tractable to be sampled and target part is also easy to be sampled with linear models. The overall algorithm pseudocode of *Augmented Flow Matching* (AFM) is sketched in Algorithm 1. And for more convenience, we also addressed the required code fragment, constructed with JAX/Flax package with python language, in Appendix D.

### 3.1 CONDITIONAL VARIANCE REDUCTION FOR FLOW MATCHING ESTIMATOR

In this section, we assume the independent CFM (I-CFM) case with $\sigma \to 0$, which is proposed and analyzed in Liu et al. (2023). According to Equation 5, the flow $u(x_t, t)$ is learned through the estimator $u(x_t, t | X_0, X_1)$, where $X_0$ is drawn from $(X_0, X_1) \sim q(x_0, x_1)$. Then, the variance of the estimator $u(x_t, t | X_0, X_1)$ is given by

$$\text{Var}_{p_0(X_0, X_1 | x_t)}(u(x_t, t) | X_0, X_1) = \mathbb{E}_{p_0(X_0, X_1 | x_t)} \left[ \|u(x_t, t) - (X_1 - X_0)\|^2 \right]. \quad (6)$$

Then, suppose that there is a random variable $Y$ which is correlated to $(X_0, X_1)$. We let $Y = (Y_0, Y_1)$ to be pairwisely concatenated to $X_0$ and $X_1$, respectively. Then, we denote $\hat{u}([X_t, Y_t], t)$ to be the augmented flow estimator, where $Y_t = (1 - t)Y_0 + tY_1$. And let the conditional flow estimator that constructs the ODE between the probability density functions of random variables $\hat{X}_0 = [X_0, Y_0]$ and $\hat{X}_1 = [X_1, Y_1]$ be given as $\hat{u}(X_t, t | Y_t)$. Applying $y_t = (1 - t)y_0 + ty_1$ with $(y_0, y_1) \sim (Y_0, Y_1)$, let $\tilde{u}(X_t, t | Y_t)$ be the flow matching estimator $u(X_t, t)$ conditioned on $Y = Y_t$. Then the conditional variance of the flow matching estimator $\tilde{u}(X_t, t | Y_t = y_t)$, defined by $u(X_t, t)$

conditioned on $Y_t = y_t$, is given by

$$\text{Var}_{p_0(X_0, X_1 | x_t, y_t)}(\tilde{u}(x_t, t | y_t) | X_0, X_1) = \mathbb{E}_{p_0(X_0, X_1 | x_t, y_t)}\left[\|\tilde{u}(x_t, t | y_t) - (X_1 - X_0)\|^2\right]. \quad (7)$$

Then the following statement on conditional variance bound implies that the variance of the flow matching estimator is mitigated when averaged over $Y$.

**Proposition 1** (The law of conditional variance). *Let the conditional variance of $u(x_t, t)$ and $\hat{u}(x_t, t | y_t)$ be defined as in Equation 6 and Equation 7. Then the variance of conditional flow on $Y = y_t$ is always less than the marginal flow.*

$$\text{Var}_{p_0(X_0, X_1 | x_t)}(u(x_t, t) | X_0, X_1) \geq \mathbb{E}_{y_t \sim Y_t}\left[\text{Var}_{p_0(X_0, X_1 | x_t, y_t)}(\tilde{u}(x_t, t | y_t) | X_0, X_1)\right]. \quad (8)$$

*Proof.* Please refer to Appendix B.1. The sketch of proof comes from the law of total variance applied to conditional distributions on $(X_0, X_1)$. ☐

This proposition shows that conditioning the flow matching loss on $Y$ about the input distribution $(X_0, X_1)$ reduces the variance of the flow matching estimator, and hence makes the training process more efficient.

Table 1: Dimension for Auxiliary variables vs. variance with Proposition 1.

| AugDim | Variance |
|--------|----------|
| 0 | 6.627 |
| 1 | 5.633 |
| 2 | 2.417 |
| 5 | 1.343 |
| 10 | 0.571 |

**Validation of Proposition 1** We take the evaluation of the flow matching estimator with the following motivating example. Let $(X_0, X_1)$ be drawn from $\mathcal{N}(-2, 1^2)$ and $\mathcal{N}(2, 1^2)$. Then we use Sampling-Importance-Resampling (SIR) to draw from $p(X_0, X_1 | x_t, y_t)$ to get $(X_0, X_1)$.

$$p_t(X_0, X_1 | x_t, y_t) = \frac{p_t(X_0, X_1 | x_t) p_t(y_t | X_0, X_1, x_t)}{p_t(y_t)} \quad (9)$$
$$\propto p_t(X_0, X_1 | x_t) p_t(y_t | X_0, X_1)$$

where $p_t(y_t | X_0, X_1, x_t) = p_t(y_t | X_0, X_1)$ since $x_t$ is deterministic with $(X_0, X_1)$. And

$$p_t(y_t | X_0, X_1) = \mathcal{N}(0.5(P_0 X_0 + P_1 X_1), 0.5^2 I), \quad (10)$$

Then, we sample from $p_t(X_0, X_1 | x_t, y_t)$ as follows.

- For $p_t(X_0, X_1 | x_t)$, we draw $(X_0, X_1)$ with Sampling-Importance-Resampling (SIR).
- For $p_t(X_0, X_1 | x_t, y_t)$, we reweight the sampled $(X_0, X_1)$ with $p_t(y_t | X_0, X_1)$.

Then, Table 1 shows that the variance of the flow matching estimator decreases with higher augmented dimensions.

We also clarify the variance reduction effect by comparing the loss convergence property. Figure 3 shows the training curve of flow matching loss with augmented random variables. For details, please refer to Section 5.1.

## 3.2 AUGMENTED FLOW MATCHING

In the previous sections, we justified how introducing the auxiliary random variable $Y$ improves the flow matching. Our interest is to improve the basic form of independent conditional flow matching (I-CFM), in which the source and target random variables in the input distribution $z \sim \pi(x_0, x_1)$ in Equation 5 is given independently, i.e., $\pi(x_0, x_1) = \pi_0(x_0)p_1(x_1)$. Then the conditional distribution is given as

$$p_t(x_t | z) = \mathcal{N}(x_t | tx_1 + (1-t)x_0, \sigma^2 I), \quad u(x_t, t | z) = x_1 - x_0, \quad (11)$$

and with $\sigma \to 0$, the marginal vector field $u(x_t, t)$ constructs the transport mapping between the distributions $\pi_0(x_0)$ and $p_1(x_1)$. As the source distribution should be endowed prior to running the ODE, $Y_0$, the fraction of $Y$ to be concatenated to the source $X_0$, should have a tractable form. In our implementation, we first assumed $Y_0 \sim \mathcal{N}(0, I)$, i.e., the standard normal distribution for simplicity that can be sampled easily with $X_0$. Then, while $Y_1 = h(X_0, X_1, Y_0)$ can have diverse forms, we took $g$ to be a linear function.

Using the augmented data, we denote the IVP-ODE with the initial condition $\hat{x}_0 = [x_0, y_0]$ as

$$\mathrm{d}\hat{x}_t = \hat{u}(\hat{x}_t, t)\mathrm{d}t, \tag{12}$$

where $\hat{x}_t = [x_t, y_t]$ is the augmented data at time $t$ and $\hat{u}$ is the corresponding the (true) augmented vector field defined in Section 3.1 that correctly constructs the ODE between $\hat{X}_0 = [X_0, Y_0]$ and $\hat{X}_1 = [X_1, Y_1]$.

With such $(Y_0, Y_1)$, we use the following augmented flow matching loss:

$$
\begin{aligned}
\mathcal{L}_{\mathrm{AFM}}(\theta) &= \mathbb{E}_{(\hat{x}_0, \hat{x}_1)}\left[\|\hat{v}_\theta(\hat{x}_t, t) - (\hat{x}_1 - \hat{x}_0)\|_2^2\right] \\
&= \mathbb{E}_{(x_0, x_1) \sim q(x_0, x_1)}\big[\|v_\theta(x_t, t) - (x_1 - x_0)\|_2^2 \\
&\quad + \mathbb{E}_{q(x_0, x_1)}\mathbb{E}_{\mathcal{N}(y_0|0, I) \times h(y_1|x_0, x_1, y_0)}\|v'_\theta(y_t, t|x_t) - (y_1 - y_0)\|_2^2\big]
\end{aligned}
\tag{13}
$$

where $x_t = (1-t)x_0 + tx_1$ (as $\sigma$ approaches to 0) and $v'_\theta(y_t, t|x_t)$ denotes the vector field conditioned on $x_t$. Our proposed method, *Augmented Flow Matching* (AFM), constructs the *auxiliary* variables $Y_0$ and $Y_1$, which are *dependent* to the original random variables $X_0 \sim \pi(x_0)$ and $X_1 \sim p_1(x_1)$. In our method, we show that the dependence between $\tilde{X}_0 = [X_0, Y_0]$ and $\tilde{X}_1 = [X_1, Y_1]$ enables efficient training and better generation quality via flow matching.

### 3.3 DESIGN CHOICE AND SAMPLING WITH AUXILIARY VARIABLE

This section shows that while any free design of $Y$ is available, even the simplest form of a single randomized linear layer is versatile. As aforementioned, we take $Y_0$ to be a standard multivariate normal random variable, since the source distribution should be easy to be sampled. For $Y_1$, when $X_0, X_1 \in \mathbb{R}^d$, We take the linear form to construct $Y_1$ as a function $h$ of $(X_0, X_1, Y_0)$, as

$$Y_1 = h(X_0, X_1, Y_0) = \lambda_1 Y_0 + \lambda_2 P(X_0 + X_1) \in \mathbb{R}^K, \tag{14}$$

where $\lambda_1, \lambda_2 \in \mathbb{R}^+$ are the hyperparameters and $P$ is a channel-wise random projection matrix. While there exists a wide range of choices of $h$, using $h$ as a learnable function, for example, requires a joint training process on learning the optimal $h$ and $\hat{v}_\theta$, which may cause the learning process using Equation 13 to be more complex. So we unifiedly fixed the hyperparameters to $\lambda_1 = \lambda_2 = 0.5$ without any arbitrary hyperparameter tuning.

- In images with RGB channels, the projection matrix $P$ functions as the $1 \times 1$ convolutional layer without bias, with 3 input channels and $K$ output channels.

- When $(X_0, X_1)$ are $D$-dimensional vectors, $P$ is defined as one dense layer without bias, which outputs $K$-dimensional vector with a single matrix multiplication.

Finally, for sampling, we first define $(x_0, y_0) \sim X_0 \times Y_0$ to be Gaussian and concatenate for the last layer. Then if

$$([x_1, y_1]) = \texttt{ODESolver}([x_0, y_0], \tilde{v}, 0, 1; \theta) \tag{15}$$

where $\texttt{ODESolver}$ is the ODE solver defined in Chen et al. (2018), then $x_1$ is the terminal solution of this augmented ODE. We provide an example in Appendix C of using auxiliary random variables $Y_t$ to be correlated or uncorrelated with $(X_0, X_1)$ with $K = 20$. The result in Figure 7 shows that the correlated auxiliary variables succeed in traversing the inner and outer circles, while the uncorrelated variables fail.

## 4 RELATED WORKS

**Normalizing Flows, Flow matching, and Rectified Flow** The lines of work in generating the differential equations, either deterministic or stochastic, have been studied actively. Rezende & Mohamed (2015); Papamakarios et al. (2021) studied the generation of complex distributions that can be mapped from the simple and tractable distributions. Even though they achieved significant progress in the autoregressive generative modeling (Chen et al., 2019; Huang et al., 2018; Papamakarios et al., 2017), these models require invertible functions to generate flows, so that heavily limits their neural architectures. Chen et al. (2018); Grathwohl et al. (2019) mitigated this issue by introducing the

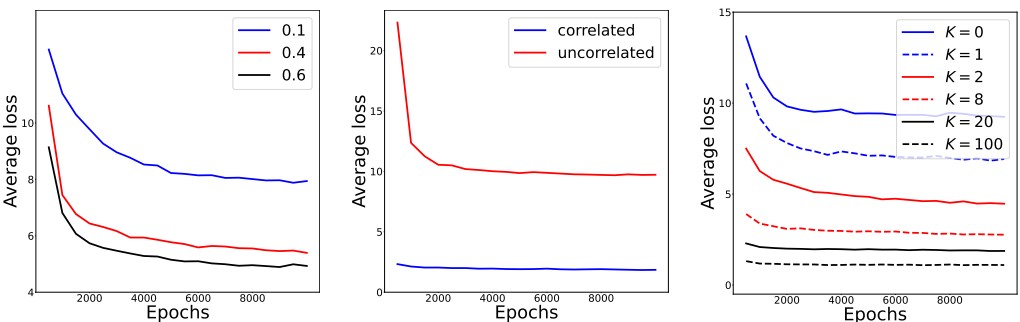

Figure 3: Flow matching loss function over training epochs. Left to right: **Left** - losses on $(\lambda_1, \lambda_2) = (0.5, \{0.1, 0.4, 0.6\})$ with $K = 2$. **Middle** - Comparison of correlated vs. uncorrelated auxiliary variable. **Right** - Comparison over various $K$, the number of auxiliary dimensions.

modeling of continuous-time function through free-form Jacobian of the vector field with memory-efficient adjoint sensitivity methods. Nevertheless, these approaches required the full simulation of the neural ODE, which is both costly and cause cumulative error caused by the ODE solver. Hence, Lipman et al. (2023); Liu et al. (2023) proposed a simulation-free approach to generate a training-efficient ODE, combining the aspects of the continuous normalizing flow and the recently emerging diffusion model (Song et al., 2021; Sohl-Dickstein et al., 2015; Ho et al., 2020).

**Regularizing ODEs with Additional Variables**   While these enabled the flexible construction of ODEs, Dupont et al. (2019) further widened the class of feasible ODEs by utilizing the latent variables that enable the path to detour by the overlapping trajectory that occurs while constructing the ODEs. Finlay et al. (2020) also augmented the continuous normalizing flow with kinetic energy term for regularization. Other lines of research interpreted augmenting additional dimensions in the dynamical systems as implementation of higher-order dynamics (Liu et al., 2021). In the diffusion model context, Dockhorn et al. (2022); Chen et al. (2024) utilized the additional dimensions of the training set as momentum or the optimal control to accelerate diffusion sampling. Some other works, such as Xu et al. (2022; 2023a), introduced the concept of Poisson field in generative modeling, and interpreted the generative modeling problem to the particle dynamics in the generalized multi-dimensional Poisson fields.

**Variance reduction in Machine Learning Methods**   Our method, AFM, lies on the line of variance reduction of the statistical estimates. Rao (1992); Blackwell (1947) showed that the variance of the estimator can be reduced by conditioning to sufficient statistics. With this, Doucet et al. (2000) used Rao-blackwellization for efficient particle filtering. Variance reduction techniques are also applied to improving diffusion models. For instance, Xu et al. (2023b); Niedoba et al. (2024) performed multiple Monte Carlo estimates of the conditional score function for single loss evaluation to reduce the training variance.

## 5  EXPERIMENTS

We clarified and validated our method, AFM, with various classes of datasets. First, we qualitatively compare AFM and the baseline flow matching method qualitatively in the single embryonic cell generation task, which deploys the fast and efficient evolution of the cell through days. Then, we confirmed our method in CIFAR-10 images quantitatively, showing that augmented FM achieves better FID. The precise description of datasets, model architecture, and training details are explained in Appendix A. For all experiments, the augmented variables $(Y_0, Y_1)$ are defined as explained in Section 3.3.

### 5.1  2D SYNTHETIC EXPERIMENT

We first give insight into the augmented flow matching with a two-dimensional synthetic experiment. The 2d synthetic distribution matching task, which is first introduced in Dupont et al. (2019), is a

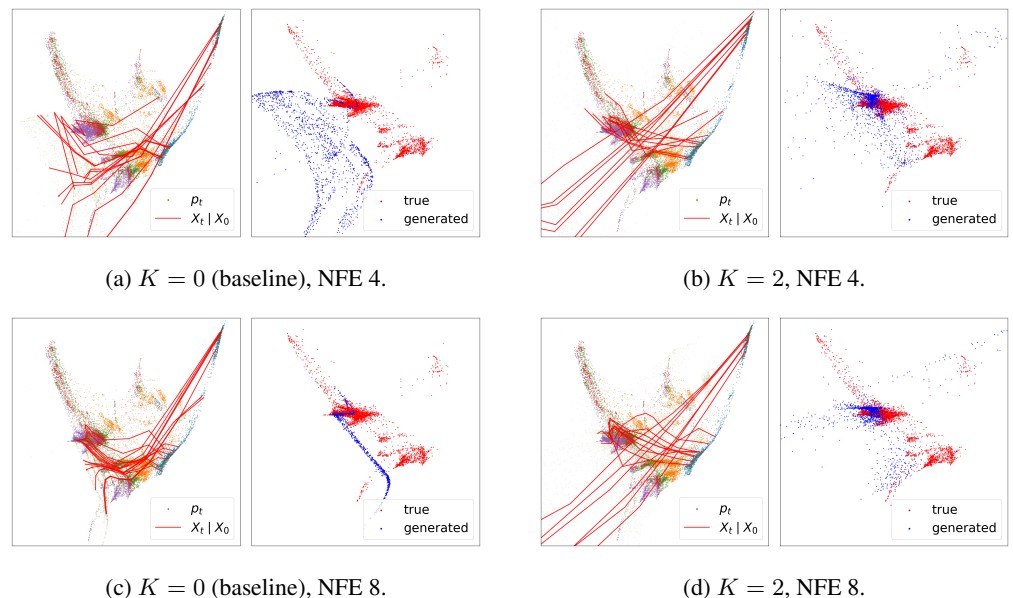

(a) $K = 0$ (baseline), NFE 4.

(b) $K = 2$, NFE 4.

(c) $K = 0$ (baseline), NFE 8.

(d) $K = 2$, NFE 8.

Figure 4: Visualization of embryonic cell evolution task. For each augmented dimension $K$ and NFE, the left and right figures represent the evolution trajectory and final points. In the left figures, the red line demonstrates the evolution of the simulated single embryonic cell from the given initial condition to the terminal point, which is pointed out in the right figures. In the **right** figures, the red and blue points stand for the features from true and generated terminal evolutions, respectively.

task of constructing a mapping between two non-overlapping distributions within a single flow. We initialize with two density functions (the foremost left figure in Figure 2) such that the **black** cloud gets out of the inner circle to the outer circle, and **red** cloud get into the inner circle from the outer circle. Each column of each row corresponds to $\Delta t = 0.1$.

Figure 2 shows that with $K = 0$, (e.g., no augmentation) one cannot fully recover the dynamics. This is reasonable because the vector fields from the black and red points overlap. On the other hand, larger $K$ implies that there are more paths to detour the conflict. Finally, when $K$ increases up to 20, one can almost fully recover the dynamics with bypassing the other point cloud family. We see that this distribution matching task is a representative example of constructing a flow that cannot be learned through a non-augmented flow matching, but can be learned from augmented flow matching.

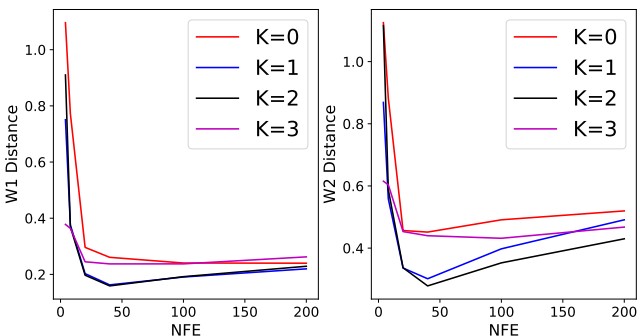

Figure 5: Wasserstein-$p$ distances (**Left**: $p = 1$, **Right**: $p = 2$) between true and generated distributions in the single embryonic cell evolution tasks.

Augmented flow matching also reduces the loss function of the conditional flow matching estimator. Figure 3 demonstrates the first two-coordinate (x-y) loss, which is the primary part of the sampling process, as other dimensions are redundant after generation finishes. Figure 3 shows that the first two-coordinate loss significantly decreases as $K$ increases. This also implies that we can achieve more efficient training with augmented dimensions.

We validated that as the number of auxiliary dimensions increases, the convergence of the flow estimator becomes faster, when applied with the same optimizer. The exception is the baseline FM,

Table 2: CIFAR-10 image generation result (in FID) with Euler and 2nd-order Heun solvers in the baseline (marked with $^\dagger$) and augmented flow matching models, evaluated with respect to the number of augmented dimensions. Every model is trained with the same model and hyperparameters, except for the first layer and the projection matrix that generates $Y_1$. The best and second best results for each NFE are highlighted with **bold** and *italic* faces, respectively. In addition, with the Heun table, we also included the computational cost, including the number of parameters and the computational cost (in terms of FLOPs) of each model.

| Method | NFE (number of function evaluations) | | | | | | | | | |
| --- | --- | --- | --- | --- | --- | --- | --- | --- | --- | --- |
| | 5 | 10 | 15 | 20 | 25 | 30 | 40 | 50 | 75 | 100 |
| Euler | | | | | | | | | | |
| AugDim=0$^\dagger$ | *38.649* | *14.235* | 9.592 | 7.647 | 6.585 | 5.910 | 5.024 | 4.571 | 3.938 | *3.625* |
| AugDim=1 | **37.435** | **13.907** | **9.415** | 7.522 | 6.491 | 5.702 | 5.056 | 4.571 | 3.976 | 3.691 |
| AugDim=2 | 39.601 | 14.478 | *9.473* | **7.421** | **6.277** | *5.564* | *4.861* | *4.412* | *3.870* | 3.637 |
| AugDim=3 | 41.912 | 15.164 | 9.640 | *7.469* | *6.280* | **5.519** | **4.789** | **4.332** | **3.820** | **3.583** |
| AugDim=4 | 39.848 | 14.620 | 9.730 | 7.674 | 6.573 | 5.863 | 5.063 | 4.569 | 4.022 | 3.746 |
| AugDim=8 | 39.738 | 14.814 | 9.839 | 7.794 | 6.693 | 5.913 | 5.149 | 4.655 | 4.021 | 3.721 |

| Method | Comp. Cost | | NFE (number of function evaluations) | | | | | |
| --- | --- | --- | --- | --- | --- | --- | --- | --- |
| | # Params | FLOPs | 10 | 20 | 30 | 40 | 50 | 60 |
| 2nd-order Heun | | | | | | | | |
| AugDim=0$^\dagger$ | 247.217 M | 8.502 G | *85.023* | 30.697 | 14.826 | 8.130 | 4.982 | 3.709 |
| AugDim=1 | + 9,223 | + 2.753 M | **81.795** | 27.296 | 13.238 | 7.298 | 4.838 | 3.717 |
| AugDim=2 | + 18,446 | + 6.586 M | 90.195 | *27.084* | **12.403** | **6.877** | **4.558** | *3.605* |
| AugDim=3 | + 27,669 | + 11.01 M | 103.227 | 29.414 | 13.871 | 7.732 | 5.219 | 4.024 |
| AugDim=4 | + 36,892 | + 16.02 M | 86.036 | **26.961** | *12.638* | *6.925* | *4.568* | **3.574** |
| AugDim=8 | + 73,784 | + 41.98 M | 90.658 | 30.870 | 14.646 | 7.768 | 4.921 | 3.688 |

$K = 0$, where the relative loss is lower than $K = 2$ or higher case. In $K = 0$ or $K = 1$ cases, the constructed flows are stuck in the local minimum, because of the topological issues on the ODE trajectory, leading the flow estimator to converge in the sub-optimal point. The variance-reducing effect is more clearly observed when $K$ becomes higher. Further analysis with relative loss is contained in Appendix C, that verifies the variance reduction property as fast convergence of the loss function.

## 5.2 IMAGE GENERATION

For quantitative analysis, we take empirical evaluation on the image generation task with CIFAR-10 datasets, compared to the independent flow matching without augmented dimensions. As mentioned in Appendix A, we used `JAX/Flax` packages on TPU v2-8 nodes. Table 2 demonstrates that the quality of images concerning FID measure has improved using both the Euler and 2nd-order Heun solver in the wide range of number of function evaluations.

Figure 1 showed an (uncurated) example image that implies the consistency of the generated image with augmented flow matching. In Figure 1, we demonstrated $64$ different images, their mean and pixel-wise standard deviations with $K = 1, 2, 3$. The image part $x_0 \in \mathbb{R}^{32 \times 32 \times 3}$ start with the same random noise, and the auxiliary part $y_0 \in \mathbb{R}^{32 \times 32 \times K}$ start with arbitrary random normal variables. The auxiliary randomness gives the stochasticity of the generated images, as shown in the image with $K = 3$ which starts from the same random variable and use deterministic Euler solver (which implies that the terminal point will be near to the initial point) but let the generated images to vary more. In addition, we also demonstrate the diversity and coverage of our generated data compared to the original data. The detailed derivation of precision and recall are introduced in (Kynkäänniemi et al., 2019).

In addition, we have reported the additional computational cost of augmented flow matching in Table 2. It shows that the amount of additional computation is negligible, costs less than $0.5\%$ of the total computational cost in both memory consumption and floating-point operations (FLOPs). This

| Method | $M = 1$ | | $M = 2$ | | $M = 5$ | |
| --- | --- | --- | --- | --- | --- | --- |
| | Pre | Rec | Pre | Rec | Pre | Rec |
| AugDim-0 (FM) | 0.499 | 0.142 | 0.628 | 0.214 | 0.796 | 0.358 |
| AugDim-1 (AFM) | 0.527 | 0.144 | 0.664 | 0.226 | 0.824 | 0.343 |
| AugDim-2 (AFM) | 0.532 | 0.156 | 0.691 | 0.241 | 0.835 | 0.363 |

Table 3: The Precision-Recall metric of CIFAR-10 dataset with 10000 generated images. $M$ stands for the hyperparameter for nearest neighbor estimation proposed in Kynkäänniemi et al. (2019).

implies that our method is extremely lightweight and easy to implement, which is ubiquitously applied to existing flow-based methods. We also report the wall-clock time of our generation in Table 4.

### 5.3 SINGLE EMBRYONIC CELL EVOLUTION

We perform our analysis with a qualitative interpretation of the evolution of single embryonic cell dynamics, proposed in Schiebinger et al. (2019). This data consists of the evolution of cells in four consecutive weeks. Then the output is projected to $D$-dimensional latent vectors for visualization. This task denotes the "distribution matching" between the initial *Day 0* data distribution (the latent vector of the initial cells) and *Day 27* data distribution (the latent vector of the final cells) (See Figure 6). If this task successfully constructs the flow between those two distributions, this implies that when the latent cell vector is initialized with *Day 0* states, then the *Day 27* state will be generated with running the ODE. The evolution trajectory and the distribution of generated terminal points with the true terminal points are demonstrated in Figure 4.

In Figure 4, we compared $K = 0$ and $K = 2$ cases, where $K$ is the number of augmented dimensions. As depicted in Figure 4, the generated terminal points with $K = 0$ case does not follow the true terminal points, while the generated points with $K = 2$ case, which utilizes two auxiliary dimensions, better follows the distribution of true points. We also measured the Wasserstein-1 (EMD) and Wasserstein-2 distance between the generated and true points in Figure 5. As AFM achieves better distribution matching, (i.e., lower Wasserstein distances) one can match the target density function more efficiently with AFM. For further results on the path on embryonic cell evolution, please refer to Appendix F.

## 6 CONCLUSION

In this paper, we analyzed some of the limitations of conditional flow matching, which is caused by the high variance of independent conditional flow matching methods. To mitigate this issue, we introduced a simple and effective method, *Augmented Flow Matching* (AFM), which correlates the training pair by concatenating auxiliary variables to them and enables variance reduction in the flow matching problem, making the training process more efficient and widening the class of ODEs to have more flexibility. Finally, with our experiment, we demonstrated the improvement in the sampling performance of the ODE constructed with the augmented variables, especially in low-NFE regimes.

**Limitations.** Although our method is able to increase the sampling efficiency of the flow-matching method, the use of the auxiliary variables result in additional parameters for the flow estimator. As shown in the experiments, the effect of the variance reduction saturates as the dimension of the auxiliary variables grows. Hence it is crucial to balance the gain from the auxiliary variables and the cost from the additional parameters.

**Ethics Statement.** While flow matching can be utilized for developing large-scale image generative models that may have negative societal impacts, our paper primarily addresses the theoretical aspects of the method. As such, the content of our paper is unlikely to cause any direct negative societal effects.

**Reproducibility Statement.** First, we have submitted the implementation of our method in the supplementary materials, to enable reproduction of our main results with the established configurations. And we also revealed the code fragments in the appendix for further research of flow-based models and related fields such as bridge matching or diffusion model researches.

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

## A    DATASETS AND ARCHITECTURES

For reproducibility and convenience, we introduce the datasets and architectures that are used in our paper.

**2d synthetic distribution matching**    For 2d synthetic distribution matching task which is depicted in Figure 2, we defined two random variables as follows.

$$X_{\text{in}} = (\varphi \cos \theta, \varphi \sin \theta), \quad \theta \sim \mathcal{U}(0, 2\pi), \varphi \sim \mathcal{N}(0, 1).$$
$$X_{\text{out}} = (\phi \cos \theta', \phi \sin \theta'), \quad \theta' \sim \mathcal{U}(0, 2\pi), \phi \sim \mathcal{N}(8, 0.5^2). \tag{16}$$

Then we define the flow matching problem by defining the source and target pairs as follows:

$$(X_0, X_1) = (X_{\text{in}}, X_{\text{out}}) \text{ with probability } p = \frac{1}{2}$$
$$(X_0, X_1) = (X_{\text{out}}, X_{\text{in}}) \text{ with probability } p = \frac{1}{2}. \tag{17}$$

Then the flow traverses the points in the inner cloud to the outer cloud, and vice versa. The architecture is MLP with two hidden layers, 64 hidden neurons with cosine and sine embedding of times in each layer, using Swish activation function. Also in this problem, we constructed the auxiliary variables $(Y_0, Y_1)$ as introduced in § 3.2.

**Single cell evolution**    The embryonic stem cell dataset is proposed by Schiebinger et al. (2019), and is accessible in `https://data.mendeley.com/datasets/hhny5ff7yj/1`. The full evolution from day 0 to 27 is demonstrated in Figure 6, starting from the right blue density clouds to the left purple clouds.

We used a neural network with 3 hidden layers, each with 64 neurons and swish activation functions. The minibatch size and the number of steps are given as $(256, 30000)$, respectively.

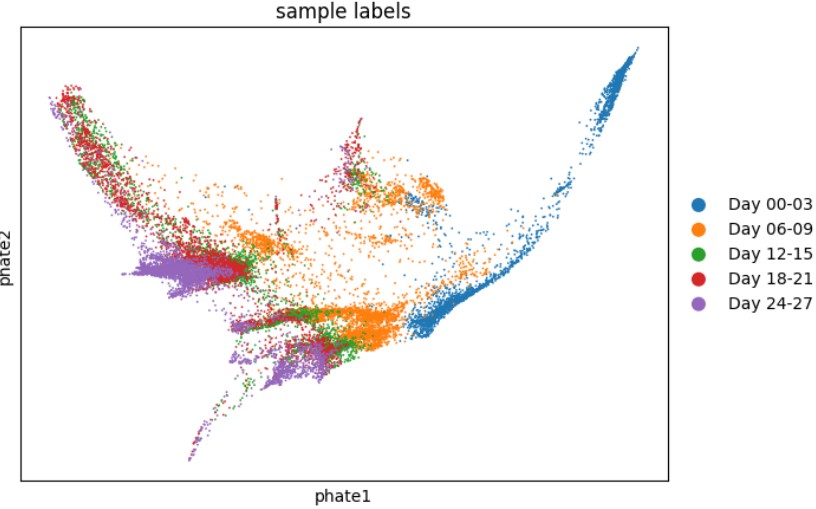

Figure 6: Single cell.

**CIFAR-10 image generation**    For CIFAR-10 image generation, we borrowed the same `NCSN++` architecture from `score_sde` repository (Song et al., 2021). Also, for all $K$, we used the same training hyperparameters. We used the `AdamW` optimizer of learning rate $2 \times 10^{-4}$ with cosine warmup of $5,000$ steps and gradient clipping. The training and sampling batch size is given as $128$ and $2,048$, respectively, as a higher batch size degraded the sampling performance and caused overfitting. The total number of iteration steps is 1.3 M, and the checkpoint with the best FID is restored, among the checkpoints that are saved every $50,000$ steps. The $50,000$ training split is used for training the model. With the v2-8 TPU, the full training costs 86 hours to run 1.3 M steps, and the sampling costs 0.4 sec per forward pass to generate $2,048$ samples.

# B    PROOFS

## B.1    PROOF OF PROPOSITION 1

**Proposition 1** (The law of conditional variance). *Let the conditional variance of $u(x_t, t)$ and $\hat{u}(x_t, t|y_t)$ be defined as in (6) and (7). Then the variance of conditional flow on $Y = y_t$ is always lower than the marginal flow, i.e.,*

$$\mathrm{Var}_{p_0(X_0, X_1|x_t)}(u(x_t, t)|X_0, X_1) \geq \mathbb{E}_{y_t \sim Y_t} \left[ \mathrm{Var}_{p_0(X_0, X_1|x_t, y_t)}(\tilde{u}(x_t, t|y_t)|X_0, X_1) \right]. \tag{18}$$

*with equality when $Y_t$ is independent to $(X_0, X_1)$.*

*Proof.* For simplicity, we let $u(x_t, t|X_0, X_1) = Z$.

$$
\begin{aligned}
\mathrm{Var}_{p_0(X_0, X_1|x_t)}(Z) &= \mathbb{E}_{p_0(X_0, X_1|x_t)} \left[ Z^2 \right] - \left( \mathbb{E}_{p_0(X_0, X_1|x_t)} [Z] \right)^2 \\
&= \mathbb{E}_Y \mathbb{E}_{p_0(X_0, X_1|x_t, Y_t)} \left[ Z^2 | Y_t \right] - \left( \mathbb{E}_{p_0(X_0, X_1|x_t)} [Z] \right)^2. \\
&= \mathbb{E}_Y \left( \mathrm{Var}_{p_0(\cdot|X_0, X_1)} [u(x_t, t)|Y_t] + \mathbb{E}_{p_0} [Z|Y_t]^2 \right) - \left( \mathbb{E}_{p_0(X_0, X_1|x_t)} [Z] \right)^2
\end{aligned}
\tag{19}
$$

As $u(x_t, t)$ conditioned on $Y_t$ is equal to $\tilde{u}(x_t, t|Y_t)$, then Equation 19 becomes

$$
\begin{aligned}
(19) &= \mathbb{E}_{Y_t} \mathbb{E}_{p_0(X_0, X_1|x_t, Y_t)} [\tilde{u}(x_t, t|Y_t)|X_0, X_1] + \left( \mathbb{E}_{Y_t} \left[ \mathbb{E}_{p_0} [u(x_t, t)|Y_t]^2 - u(x_t, t)|Y_t \right] \right) \\
&= \mathbb{E}_{Y_t} \mathbb{E}_{p_0(X_0, X_1|x_t, Y_t)} [\tilde{u}(x_t, t|Y_t)|X_0, X_1] + \mathrm{Var}_{Y_t} \mathbb{E}_{p_0(X_0, X_1|x_t, Y_t)} [u(x_t, t)|Y_t] \\
&\geq \mathbb{E}_{Y_t} \mathbb{E}_{p_0(X_0, X_1|x_t, Y_t)} [\tilde{u}(x_t, t|Y_t)|X_0, X_1],
\end{aligned}
\tag{20}
$$

with equality condition when $\mathrm{Var}_{Y_t} \mathbb{E}_{p_0(X_0, X_1|x_t, Y)} [u(x_t, t)|Y] = 0$. This includes the case of independent conditional flow matching case described in Liu et al. (2023), where $Y_t$ is an empty random variable.    □

# C    ADDITIONAL RESULTS

**Effect of correlation of auxiliary variable $Y$ with data variables $(X_0, X_1)$**    This part provides additional rationale that supports the usage of correlated auxiliary variables in Section 3.3. With the same setting as Figure 2 with $K = 20$, we validated the effect of correlation between the source and target variables $(X_0, X_1)$ and the augmented variable $Y_t$. According to Figure 7, using correlated auxiliary random variables is more beneficial, succeeding in constructing the path between the source and target distributions.

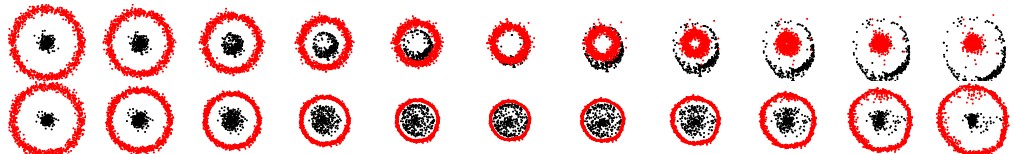

Figure 7: **Up**: correlated ($\lambda_1 = \lambda_2 = 0.5$, **Down**: uncorrelated ($Y$ is independent to $(X_0, X_1)$.) with $K = 20$.

**Fast convergence of relative loss**    For fair comparison, we observe the convergence properties of the loss function in Figure 8, which is the modification of Figure 3 where the relative loss compared to the converging checkpoint rather than the absolute loss is use, to verify the variance reduction property. In the **left** figure, we fix the $\lambda_1$ coefficient, which determines the randomness of the starting point, and varied $\lambda_2$ coefficient, which stands for the correlation between $X_1$ and $Y_t$. Not only the absolute loss (as shown in Figure 3, but also the relative loss converges faster if $\lambda_2$ is higher, i.e., the random variable is more correlated.

In the **middle** figure shows that using the uncorrelated auxiliary variable ($\lambda_1 = \lambda_2 = 0.0$) has much slower converging property than correlated ($\lambda_1 = \lambda_2 = 0.5$) case. Last, **right** figure compares the convergence property for various $K$. This shows that with higher $K$, the relative loss converges faster, which implies the variance reduction property holds.

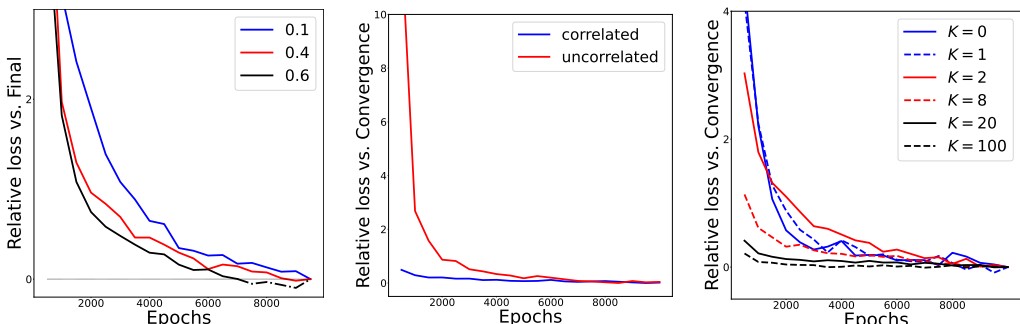

Figure 8: Left to right: **Left** - losses on $(\lambda_1, \lambda_2) = (0.5, \{0.1, 0.4, 0.6\})$ with $K = 2$. **Middle** - Comparison of correlated vs. uncorrelated auxiliary variable. **Right** - Revised Figure 3 with relative loss vs. convergence.

**Wall-clock time of CIFAR-10 generation**  In addition to the cost in terms of FLOPs and memory in Table 2, we also report the wall-clock time of our CIFAR-10 generation in Table 4. We used a single TPU v3-8 node and used $2,048$ minibatch size.

| Method | Wall-clock time | Time per image per NFE |
|---|---|---|
| AugDim-0 (FM) | 39.944 s | 0.813 ms |
| AugDim-1 (AFM) | 39.366 s | 0.801 ms |
| AugDim-2 (AFM) | 39.448 s | 0.802 ms |

Table 4: The wall-clock time of our CIFAR-10 generation.

# D PYTHON CODE FOR AFM

Augmented flow matching, especially in a linear form, is extremely easy to implement, with just a few line of codes. We provide the fragment of codes, written by `JAX/Flax`.

## D.1 DEFINE THE AFM MODULE.

```python
import flax.linen as nn
import jax
import jax.numpy as jnp
import functools

aug_dim = 10 # Modify this.
seed = 9999  # Modify this.
rng = jax.random.PRNGKey(seed)
DenseLayer = nn.Dense(aug_dim, use_bias=False)
rng, step_rng = jax.random.split(rng)
kernel_arr = jax.random.normal(step_rng, shape=(n_channels, aug_dim))
afm_fn = functools.partial(afm_model.apply, {'params': {'kernel':
    kernel_arr}})
```

## D.2 USE AFM MODULE TO PERTURB DATA

```python
def afm_data(rng, x0, x1, aug_dim, afm_fn, lambda1=0.5, lambda2=0.5):
    """
    Input
      x0: data drawn from source distribution.
      x1: data drawn from target distribution.
      aug_dim: augmented dimension.
      afm_fn: AFM function.
      lambda1, lambda2: weights
    Return
      y0: augmented source data. Should be easily sampled.
      y1: augmented target data. Function output of y0, x0, x1.
    """
    rng, step_rng = jax.random.split(rng)
    y0 = jax.random.normal(step_rng, shape=x0.shape[:-1] + (aug_dim,))
    y1 = lambda1 * y0 + lambda2 * afm_fn(x0 + x1)
    y0 = jnp.concatenate([x0, y0], axis=-1)
    y1 = jnp.concatenate([x1, y1], axis=-1)
    return y0, y1
```

# E    MORE UNCURATED IMAGE EXAMPLES

In this section, we cover more image examples generated by AFM.

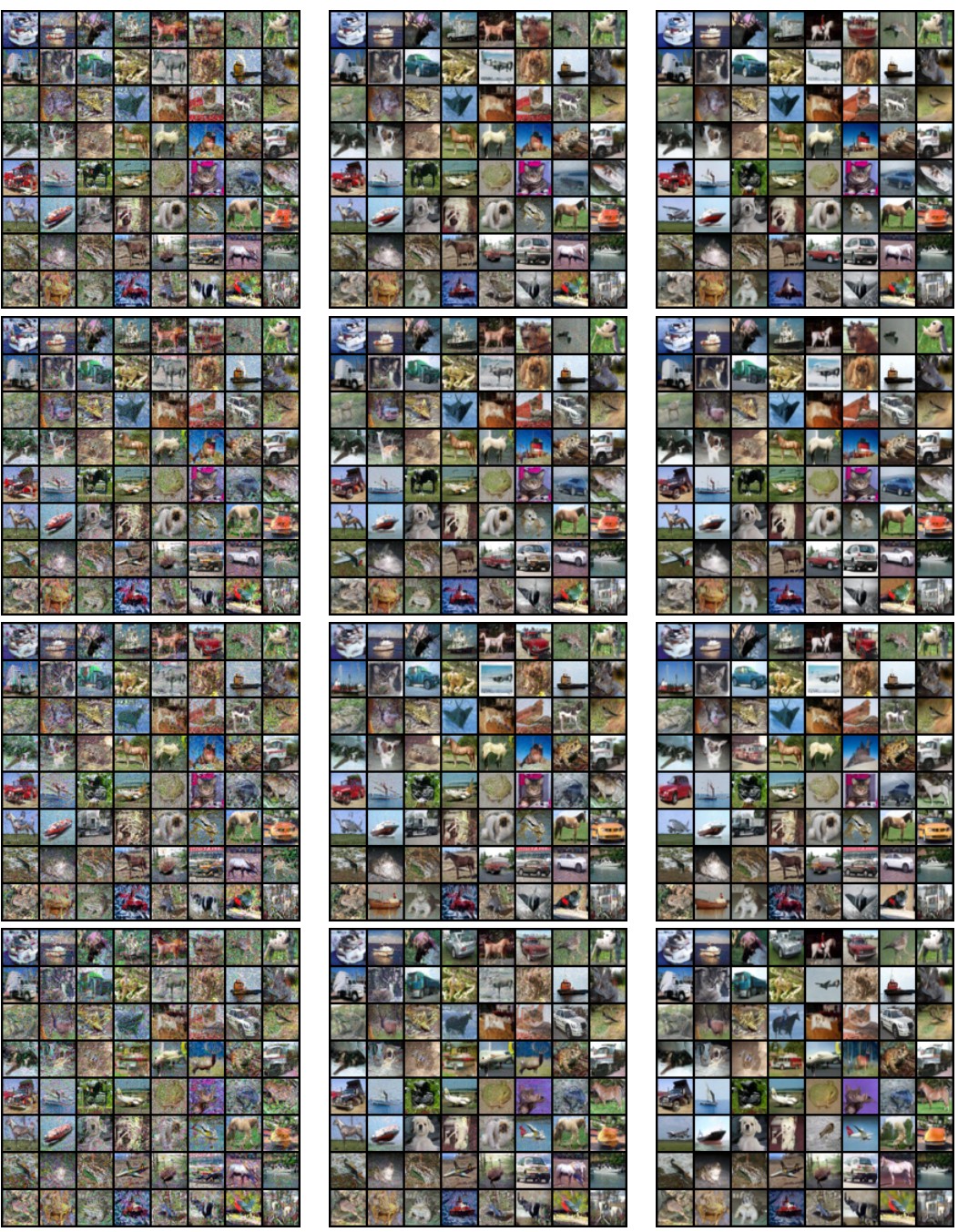

Figure 9: Up→Down: Augmented dimension $\{0, 1, 2, 3\}$. Left→Right: NFE $\{10, 20, 50\}$ with $2^{\text{nd}}$-order Heun solver.

## F MORE RESULTS ON EMBRYONIC CELL EVOLUTIONS.

In this section, we report extensive results of Figure 4 on various auxiliary dimensions $K$ and number of function evaluations (NFE). We find that in the high NFE case (NFE $= 400$), the generated data with baseline ($K = 0$) is limited to the low-dimensional manifold, while $K = 1$ and $K = 2$ succeed to find diverse cells.

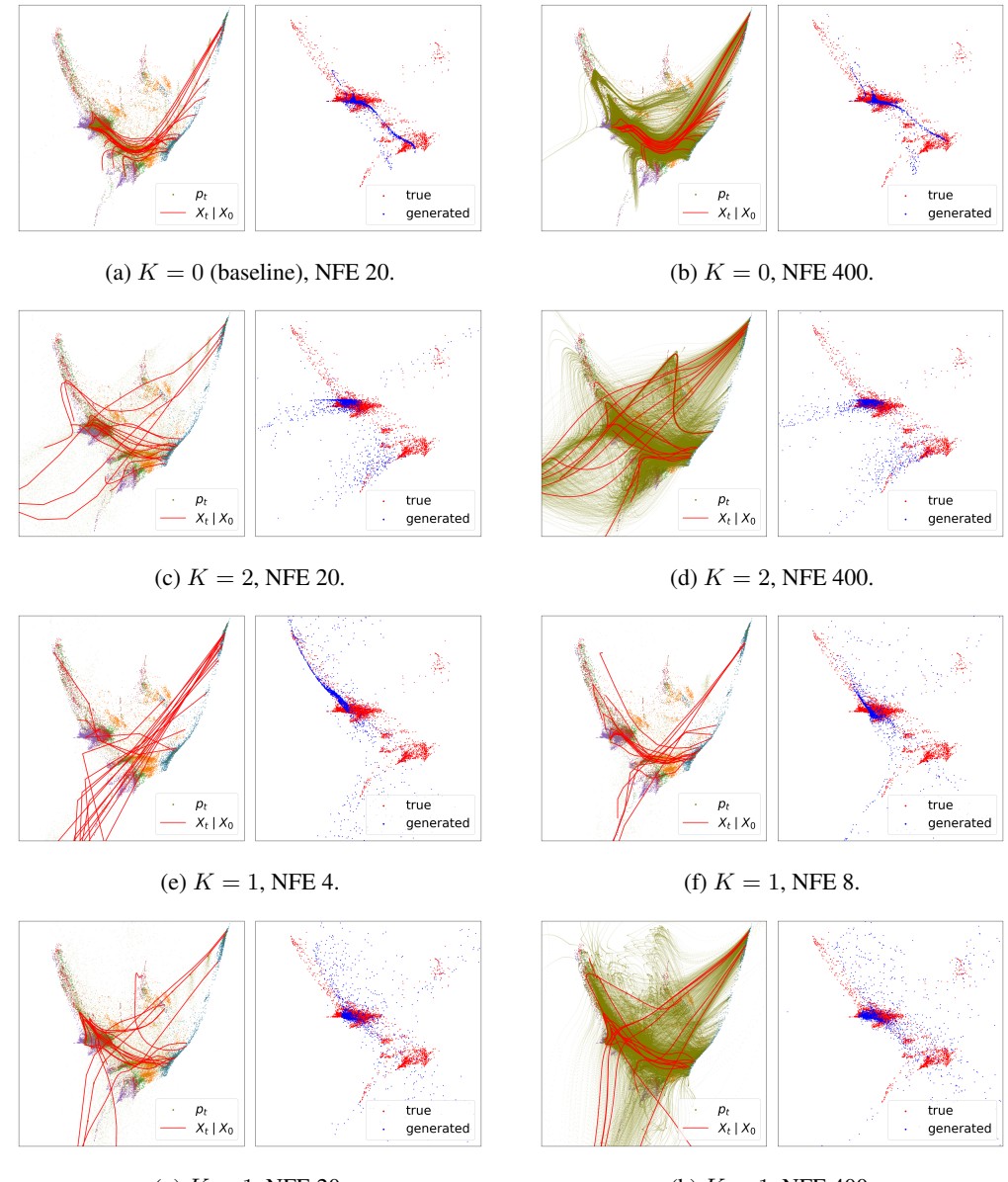

(a) $K = 0$ (baseline), NFE 20.

(b) $K = 0$, NFE 400.

(c) $K = 2$, NFE 20.

(d) $K = 2$, NFE 400.

(e) $K = 1$, NFE 4.

(f) $K = 1$, NFE 8.

(g) $K = 1$, NFE 20.

(h) $K = 1$, NFE 400.

