# OpenReview forum: "Augmented Flow Matching via Variance Reduction with Auxiliary Variables"
_ICLR.cc/2025/Conference — Submitted to ICLR 2025_

### Official Review · Reviewer_KoFb · 2024-10-19

**Soundness:** 3
**Presentation:** 3
**Contribution:** 2
**Rating:** 6
**Confidence:** 3

**Summary:**

This paper proposes an augmented flow matching (AFM) framework, which reduces the conditional variance by introducing auxiliary random variable correlated to training pair (use linear combination for simplicity in this paper). After justifying the claim and validating the proposed method by 2D synthetic data, they applied their method to single embryonic cell evolution and CIFAR-10 dataset.

**Strengths:**

1. The paper is well written and the idea is clear.
2. The modification is computational light.
3. The variance reduction strategy is simple and general, and it can be easily adapted to different situations by different design of $Y$.

**Weaknesses:**

1. The major concern is mentioned in the discussion
2. A typo? line 205, "where $X_0$ is drawn..." should be "where $(X_0, X_1)$ is drawn..."

**Questions:**

No major questions.

---

### Official Review · Reviewer_YEJm · 2024-11-03

**Soundness:** 3
**Presentation:** 3
**Contribution:** 3
**Rating:** 5
**Confidence:** 4

**Summary:**

The paper addresses the task of flow matching for generating an ODE such that its path traverses between two distributions. To reduce variance during training, the paper proposes an augmented flow matching framework that introduces auxiliary variables that are correlated to the training pair. The paper provides some empirical results to demonstrate that the proposed training procedure leads to improved performance.

**Strengths:**

S1. The paper proposes a novel idea that is very simple yet effective. It is easy to implement and has a relatively low computational and memory overhead. There is the potential, albeit unexplored, to combine the approach with other flow matching improvement techniques.

S2. The motivation for the proposed technique is well-developed, with useful examples to aid understanding and support the intuition.

S3. The paper provides experimental results for synthetic data, as well as image and cell evolution data.

**Weaknesses:**

W1. While there is no need for a technique to be unnecessarily complicated, and a paper should not be penalized for proposing a simple method, especially when it is effective (there is elegance and robustness in simplicity), the technical contribution of the paper is limited. There are several important issues that are not resolved. Perhaps most importantly, the experimental results clearly indicate the importance of the choice of the augmentation dimension and the number of function evaluations. The proposed technique even makes things worse (at a slightly higher computational and memory cost) if there is not a judicious selections. The paper would be considerably stronger if it investigated how to choose these parameters and proposed an effective strategy.

W2. The experiments provide some support for the claim that the proposed method leads to improved performance, but the experimental analysis is not convincing or thorough. The paper does not report any measure of the variability in the experiments. It is not clear to me if these are the results from a single run or the average over multiple runs. There is no attempt at statistical significance testing to establish that the performance differences are meaningful. This is a particular concern when there is inconsistent behaviour. For example, in Table 2 Heun, the performance for AugDim=3 is consistently worse for all NFE than both AugDim=2 and AugDim=4. This doesn’t really make sense. It raises concerns about the consistency of the results and the experimental assessment.

W3. The paper claims that “This approach can be plugged-in to other existing training methods that enhances efficiency, such as optimal transport or curvature-minimizing approach.” While it may be true that it can be combined with these approaches, the paper provides no evidence that the combination is useful. In general, the paper does not compare experimentally to any other methods that are designed to improve the performance of flow matching beyond basic independent sampling. It would at least be useful to understand whether the technique, when employed on CIFAR-10, for example, outperforms or approaches the performance of some of the much more computationally burdensome strategies. For example, how does it compare to ReFlow?

W4. The discussion for the cell evolution experiment could be improved. Qualitatively, the distributions obtained using the augmented flow matching approach are better, but they are still very different from the original. The discussion is limited to a statement that the proposed approach leads to a collection of samples that “better follows the distribution of true points”. Are they good enough? How can one tell?

**Questions:**

Q1. How can one choose the number of evaluations and the augmentation dimension?

Q2. Please explain if the experimental results are derived from multiple runs. If so, please characterize the variability in performance. Please conduct statistical significance tests where appropriate to establish that there is a meaningful difference between the baseline and the proposed approach.

Q3. Please comment on other improvement approaches. (i) Please compare the achieved performance to the previous methods (with the understanding that they are much more computationally heavy); and (ii) please explain whether there is evidence that the adoption of the proposed method in combination with these improvement strategies leads to improvement. If not, then the claim in the paper should be adjusted to acknowledge that there is potential, but it is not clear yet if the combination is useful. This avenue is particularly intriguing for me, since there have been recent proposals for simplified versions of rectified flow matching, for example, which might be computationally reasonable while retaining effectiveness. If the proposed strategy leads to further improvement, then this would be most welcome.

Q4. Please provide a more detailed discussion of the cell evolution experiment. Why does the Wasserstein distance increase as NFE increases in Figure 5? This behaviour doesn’t appear to be evident for CIFAR-10, where the metrics are effectively monotonically decreasing with NFE.

---

### Official Review · Reviewer_G1Cd · 2024-11-04

**Soundness:** 2
**Presentation:** 3
**Contribution:** 2
**Rating:** 1
**Confidence:** 5

**Summary:**

The paper studies the use of augmented variables for improving flow matching. The augmented variables change the reference probability path, and make it so the variance of the conditional estimator (eq 6) is lower (prop 1): my understanding here is that there are not as many overlapping paths in the augmented space. Algorithm 1 shows the proposed way of augmenting the path, and S3.3 discusses some design choices on how to choose the path. The experimental results apply this to synthetic 2D flows (S5.1), CIFAR-10 generation (S5.2), and embryonic cell evolution (S5.3)

**Strengths:**

The idea of augmenting a flow with additional variables is appealing as a way of changing the reference path so it's hopefully easier to learn or integrate.

**Weaknesses:**

While I do agree with the motivation for the paper, I have marked it for a strong reject because I feel that the experimental results are insufficient in the current form. I am very open to discussing these through the rest of the review period.

1. My biggest concern is that the experimental results do not improve upon the best-known flow matching results in any setting, and do not adequately compare to other flow matching variations. This is concerning as the idea and implementation of augmented variables is straightforward and easy to try in every flow matching setting and application. Concretely:
    + **1a.** In CIFAR-10 modeling: rectified flow reports an FID of 2.58 in comparison to the submitted paper's best FID of ~3.5 --- these comparisons are omitted in the submitted paper. The experimental results of the paper indicate/argue that the augmentations are helpful for lower NFE, so it seems fair to compare to other flow matching modifications that also create straighter paths, such as rectified flows evaluated with the Euler/2nd-order Huen methods, or mini-batch or multisample flow matching.
    + **1b.** Flows and transport on the embryonic cell evolution on the Waddington OT dataset by Schiebinger et al. (2019) have been extensively studied, and cited almost 1000 times. This dataset/setting is used in Figure 5 and other parts of the submitted paper, but do not compare to or reference any previously published results on this dataset. This makes it extremely difficult to assess the experimental comparison.
2. I have another minor concern that the augmented variables could hurt the performance by taking away modeling capacity. I believe this is why they carefully select the number of augmenting variables to only be a few, as likely the performance is significantly hurt otherwise, 2) Table 2 stops the NFE at 100 steps for Euler and 60 steps for the 2nd-order Huen. These seem chosen at exactly the NFE where the effect of the flows with the augmented variables disappears, so it's possible the FID is hurt when integrating with a highly accurate ODE solver.

**Questions:**

I do not have any further specific questions, and am open to discussing the weaknesses above

---

### Official Review · Reviewer_AALw · 2024-11-04

**Soundness:** 2
**Presentation:** 2
**Contribution:** 2
**Rating:** 3
**Confidence:** 4

**Summary:**

This paper proposes a novel method to reduce the variance of flow-matching loss in ODE-based generative models. The authors show both theoretically and empirically that adding  auxiliary variables that are correlated to the training data reduces the variance of the target. Based on this, they propose to construct the auxiliary variables through a random projection of the training data. Experimental results confirm the effectiveness of their proposed method.

**Strengths:**

The investigated topic is interesting. The paper is well motivated. The proposed method is easy to implement.

**Weaknesses:**

1. It's unclear how much the proposed method can reduce the variance when the data distribution is different.
2. It requires more demonstration on the robustness of the proposed algorithm with respect to the choice of random projection matrix P.
3. The effect of the proposed method highly depends on the number of auxiliary dimensions, but it lacks a criterion to determine it beforehand.
4. The arrangement of figures and tables looks messy.
5. The notations in Section 3 needs further clarification.

**Questions:**

1. How much the variance can be reduced by the proposed method?
2. Can the authors provide some criteria for choosing an appropriate augmented dimension K?
3. Can the authors explain more on the results shown in Table 3?
4. In Table 2, the result with 2nd-order Heun solver and AugDim=3 seems abnormally bad, can the authors explain the reason?

---

### Meta-Review · Area_Chair_ZcZh · 2024-12-19

**Metareview:**

The reviewers of this paper recognize the importance and timeliness of the problem addressed. However, they found limited improvement in the proposed algorithm and noted a lack of sufficient novelty. Reviewer YEJm articulated this point clearly in their review, stating that if the performance of the simple algorithm had been stronger, its simplicity and robustness could have provided additional merit for publication. Furthermore, the authors did not engage with the reviewers in the rebuttal period, leaving the key questions unaddressed. I encourage the authors to explore alternative methods of analyzing and validating their algorithm, as this could make a significant difference moving forward.

**Additional Comments On Reviewer Discussion:**

There were none.

---

### Decision · Program_Chairs · 2025-01-22

Reject